# Elevated Flotillin-1 in Saliva and Salivary Glands: A Novel Non-Invasive Biomarker in an Alzheimer’s Disease Mouse Model

**DOI:** 10.3390/diagnostics16010061

**Published:** 2025-12-24

**Authors:** Sunao Kawakami, Cha-Gyun Jung, Rieko Inoue, Tomohisa Nakamura, Soh Sato, Makoto Michikawa

**Affiliations:** 1Periodontology, The Nippon Dental University Graduate School of Life Dentistry at Niigata, Niigata 951-8580, Japan; sunao.kawakami@ngt.ndu.ac.jp; 2Center for Nursing International Promotion, Graduate School of Medical Sciences, Nagoya City University, 1 Kawasumi, Mizuho-cho, Mizuho-ku, Nagoya 467-8601, Japan; jung@med.nagoya-cu.ac.jp; 3Department of Biochemistry, Graduate School of Medical Sciences, Nagoya City University, 1 Kawasumi, Mizuho-cho, Mizuho-ku, Nagoya 467-8601, Japan; 4Oral and Maxillofacial Surgery, Nagoya City University East Medical Center, 1-2-23 Wakamizu, Chikusa-ku, Nagoya 464-8547, Japan; 5Department of Maxillofacial Surgery, Graduate School of Medical Sciences, Nagoya City University, 1 Kawasumi, Mizuho-cho, Mizuho-ku, Nagoya 467-8601, Japan; 6Department of Periodontology, The Nippon Dental University School of Life Dentistry at Niigata, 1-8 Hamaura-cho, Chuo-ku, Niigata 951-8580, Japan; 7Department of Neurophysiology and Brain Science, Graduate School of Medical Sciences, Nagoya City University, 1 Kawasumi, Mizuho-cho, Mizuho-ku, Nagoya 467-8601, Japan

**Keywords:** flotillin-1, Alzheimer’s disease, biomarker, saliva, Aβ_42_

## Abstract

**Background/Objectives:** Alzheimer’s disease (AD) is currently diagnosed using established biomarkers, such as reduced cerebrospinal fluid (CSF) Aβ_42_, increased phosphorylated tau, and cerebral amyloid levels detected by PiB-PET. Because these methods are invasive or require specialized facilities, less invasive and easily detectable biomarkers are needed. Flotillin-1 concentrations are reduced in the CSF and serum of patients with AD. This study examined whether flotillin-1 in saliva, a less invasive specimen than blood, could serve as a biomarker. **Methods:** Wild-type (WT) and *App^NL–G–F^* (APP knock-in; APP-KI) mice were used to create four groups (2 and 9 months of age, six animals per group). Saliva and salivary glands were collected, and flotillin-1 levels were measured using Western blotting. Intracellular signaling pathways regulating flotillin-1 and salivary gland Aβ_42_ levels were analyzed using Western blotting and ELISA, respectively. **Results:** Flotillin-1 levels in the saliva and salivary glands were significantly higher in the 9-month-old APP-KI group than in all other groups, including age-matched WT mice. Phosphorylated extracellular signal-regulated kinase (p-ERK) levels were also significantly elevated in the 9-month-old APP-KI group, whereas phosphorylated c-Jun N-terminal kinase (p-JNK) levels did not differ significantly. Salivary gland Aβ_42_ levels were markedly increased only in the 9-month-old APP-KI group. **Conclusions:** Flotillin-1 levels in saliva and salivary glands were significantly elevated in the presence of AD pathology. Aβ accumulation in the salivary glands likely activates the ERK signaling cascade, promoting flotillin-1 expression and secretion. Thus, salivary flotillin-1 may serve as a promising noninvasive biomarker for the early diagnosis of Alzheimer’s disease.

## 1. Introduction

Dementia is defined as “an acquired syndrome characterized by a decline from a previous level in at least two of the following cognitive domains: memory, executive function, attention, language, social cognition and judgment, psychomotor speed, visual perception, or visuospatial cognition” [1]. With the global population aging, the number of patients with dementia is increasing rapidly; as of 2020, an estimated 50 million people were affected worldwide, with this number projected to rise to 130 million by 2050 [2]. The major types of dementia include Alzheimer’s disease (AD), vascular dementia, dementia with Lewy bodies, and frontotemporal dementia, with AD accounting for 60% or more of all cases [3].

AD is characterized by cerebral amyloid-β (Aβ) deposition (amyloidosis) and neurofibrillary tangles (tauopathy). Aβ is a peptide generated through the processing of amyloid precursor protein (APP). Under physiological conditions, Aβ is rapidly metabolized and cleared; however, increased production, impaired transport, or reduced degradation can lead to Aβ accumulation in the brain, resulting in plaque formation. Aβ aggregates disrupt synaptic function and promote abnormal tau phosphorylation, resulting in neurofibrillary tangle formation. These tangles induce intracellular transport defects, ultimately leading to neuronal death. The progressive loss of neurons contributes to brain atrophy, which underlies the onset and progression of AD [4].

The differential diagnosis of AD relies on a combination of clinical symptom assessment, neuropsychological testing, neuroimaging with magnetic resonance imaging (MRI) and positron emission tomography (PET), as well as the measurement of serum and cerebrospinal fluid (CSF) biomarkers. Cerebral Aβ accumulation begins before the onset of clinically detectable cognitive impairment [5]. Accordingly, biomarkers that reflect brain Aβ deposition are critical for the early diagnosis of AD. Particularly, decreased CSF Aβ_1–42_ combined with elevated phosphorylated tau levels is considered, alongside PET imaging with Pittsburgh compound B (PiB-PET), to be one of the most reliable biomarkers [6]. However, CSF testing requires a lumbar puncture, which is invasive and burdensome for patients. PiB-PET, meanwhile, involves radiation exposure, requires advanced expertise, and depends on large-scale facilities. Thus, although current biomarkers offer excellent diagnostic accuracy, their invasiveness and technical demands represent major barriers to screening and diagnosis before symptom onset. These limitations underscore the need to identify biomarkers in specimens that can be obtained easily and noninvasively.

Saliva represents a noninvasive specimen available for biomarker discovery. It is an extracellular fluid secreted by the salivary glands and plays important physiological roles in digestion, antimicrobial defense, and oral lubrication. Recent reports suggest that AD may affect the brain and peripheral organs, such as the autonomic nervous system and salivary glands. Patients with AD exhibit significant salivary gland dysfunction, with decreased resting and stimulated saliva secretion from the submandibular and parotid glands [7,8]. Thus, AD may influence salivary volume, the molecular composition, and secretion dynamics of saliva. Saliva, therefore, is a promising, easily obtainable specimen for AD biomarker exploration. Aβ has been detected in the saliva and salivary glands of AD model mice, suggesting its potential as a salivary biomarker [9]. However, because salivary Aβ levels are lower and more variable than those in CSF, identifying salivary biomarkers with greater accuracy and reproducibility is necessary.

When considering saliva diagnostic markers, we considered flotillin-1 to be a candidate protein. Because we had found that flotillin-1 levels in the serum and CSF of patients with AD are lower than those in healthy controls and show an inverse correlation with cerebral amyloid deposition, supporting its potential as a biomarker for AD [10,11]. Moreover, it has been reported that flotillin-1 regulates the intracellular transport and processing of APP, thereby contributing to Aβ aggregation [12,13]. However, it remains unclear whether flotillin-1 levels in saliva, an even less invasive specimen, undergo changes in the context of AD pathologies. Previous studies have demonstrated that the MAPK pathways regulate flotillin-1 expression in several cell types, including neurons, macrophages, and epithelial cells. In these cells, Aβ-induced activation of the JNK and ERK pathways modulates flotillin-1 expression and membrane microdomain dynamics. Because JNK is primarily involved in stress- and inflammation-related signaling, and ERK plays a key role in cell survival and membrane trafficking, we also examined JNK/ERK activation and Aβ levels in the salivary glands in addition to measuring salivary and glandular flotillin-1 levels.

In this study, we investigated how flotillin-1 levels in saliva and salivary glands change under AD pathologies, as well as the mechanisms regulating these changes, using an AD mouse model.

## 2. Materials and Methods

### 2.1. Animals

Male mice were divided into four groups: wild-type (C57BL/6J; WT) and *App^NL–G–F^* (amyloid precursor protein knock-in; APP-KI) mice, each at 2 and 9 months of age. Each group comprised six animals (total *n* = 24). APP-KI mice, obtained from the RIKEN BioResource Research Center (RIKEN BRC, Tsukuba, Japan), exhibit progressive age-dependent cerebral Aβ deposition [14]. WT mice were purchased from Japan SLC (Hamamatsu, Japan). All mice were housed in the same facility under a 12 h light/dark cycle with ad libitum access to food and water. All procedures were approved by the Animal Care and Use Committee of Nagoya City University (protocol no. 21-029).

### 2.2. Sample Collection and Measurement of Salivary Secretion

Anesthesia was induced by intraperitoneal injection of a mixed anesthetic comprising medetomidine hydrochloride (Kyoritsu Seiyaku Corporation, Tokyo, Japan; 1 mg/mL), midazolam (Nichi-Iko Pharmaceutical Co., Ltd., Gifu, Japan; 5 mg/mL), and butorphanol tartrate (Meiji Co., Ltd., Kumamoto, Japan; 5 mg/mL) at a dose of 10 mL/kg. To stimulate salivary secretion, pilocarpine hydrochloride (FUJIFILM Wako Pure Chemical Corporation, Osaka, Japan; 0.05 mg/mL) was administered intraperitoneally at 10 mL/kg. The mice were positioned prone, and whole saliva was collected from the oral cavity using a micropipette over a 30 min period. Following saliva collection, the mice were transcardially perfused with phosphate-buffered saline (PBS; 137 mM NaCl, 2.7 mM KCl, 8.1 mM Na_2_HPO_4_, 1.5 mM KH_2_PO_4_; pH 7.4), and the salivary glands were dissected immediately. All samples were promptly stored at −80 °C. Salivary secretion volume was determined by measuring the difference in the mass of a 1.5 mL tube before and after saliva collection.

### 2.3. Total Protein Concentration in Saliva and Salivary Glands

To analyze flotillin-1 levels in saliva, the salivary secretion volume and total protein concentration were evaluated. Salivary protein concentrations were measured using a Qubit 4 fluorometer (Thermo Fisher Scientific, Waltham, MA, USA). Salivary glands were homogenized in RIPA buffer (50 mM Tris-HCl, pH 7.6; 150 mM NaCl; 1% NP-40; 0.5% sodium deoxycholate; 0.1% SDS) supplemented with protease (Roche Diagnostics GmbH, Mannheim, Germany) and phosphatase inhibitors (FUJIFILM Wako Pure Chemical Corporation, Tokyo, Japan). Protein concentrations in the salivary gland extracts were measured using a BCA protein assay kit (Thermo Fisher Scientific, Waltham, MA, USA).

### 2.4. Western Blot Analysis

Equal amounts of protein were separated using SDS-polyacrylamide gel electrophoresis (SDS-PAGE), and transferred onto polyvinylidene difluoride (PVDF) membranes (Millipore, Billerica, MA, USA). Membranes were blocked in TBS-T containing 5% skim milk and incubated overnight at 4 °C with the following primary antibodies: anti-phospho-p44/42 MAPK (Erk1/2, Thr202/Tyr204, #9106; Cell Signaling Technology, Inc., Danvers, MA, USA), anti-phospho-SAPK/JNK (Thr183/Tyr185, #9255; Cell Signaling Technology, Inc.), anti- Flotillin-1 (sc-133153; Santa Cruz Biotechnology, Inc., Dallas, TX, USA), and anti-β-actin (Cat. No. 20536-1-AP; Proteintech Group, Inc., Rosemont, IL, USA) antibodies. The membranes were incubated for 1 h with species-specific HRP-conjugated secondary antibodies. The following secondary antibodies were used: Anti-mouse IgG HRP-linked Antibody (Cell Signaling Technology, Danvers, MA, USA) and Anti-rabbit IgG HRP-linked Antibody (Cell Signaling Technology, Danvers, MA, USA). Immunoreactive bands were visualized using ImmunoStar Zeta or ImmunoStar LD (FUJIFILM Wako Pure Chemical Corporation). Chemiluminescent signals were detected using an Amersham Imager 680 (GE Healthcare, Marlborough, MA, USA). Western blot data were analyzed using ImageJ software (National Institutes of Health, Bethesda, MD, USA; version 1.54p). β-actin was used as an internal control (Beta Actin Polyclonal Antibody, Cat. No. 20536-1-AP; Proteintech Group, Inc., Rosemont, IL, USA).

### 2.5. Aβ Enzyme-Linked Immunosorbent Assay (ELISA)

To investigate factors underlying changes in intracellular signaling within the salivary glands, Aβ_42_ levels were analyzed in the salivary glands. Measurements were performed using a Human Aβ_42_ ELISA kit (Wako Pure Chemical Industries, Osaka, Japan). Samples were prepared according to the manufacturer’s protocol, and absorbance was measured at 450 nm.

### 2.6. Statistical Analysis

Data are presented as mean ± standard deviation (SD). Normality was assessed using the Shapiro–Wilk test. The Kruskal–Wallis test was used for group comparisons, and the Steel–Dwass test was applied for multiple comparisons. A *p*-value of <0.05 was considered statistically significant. Statistical analyses were performed using Bell Curve for Excel, version 4.05 (Social Survey Research Information Co., Ltd. [SSRI], Tokyo, Japan).

## 3. Results

First, we evaluated the salivary secretion volume and total protein concentration in all groups. No significant differences were found in salivary secretion volume between APP-KI and WT mice at either 2 or 9 months of age (*p* > 0.05); similarly, no significant differences were observed in total protein concentrations in saliva (*p* > 0.05) (Appendix A).

Next, we investigated whether flotillin-1 levels in the saliva and salivary gland can be affected by age and AD pathologies using Western blotting. Western blot analysis showed that the 9-month-old APP-KI group had significantly higher levels of salivary flotillin-1 than the 2-month-old APP-KI and WT groups, as well as the 9-month-old WT group (Figure 1).

Salivary gland flotillin-1 levels were also significantly higher in the 9-month-old APP-KI group than in the 2-month-old APP-KI and WT groups, as well as the 9-month-old WT group. Moreover, the 9-month-old WT group showed significantly higher levels than the 2-month-old WT group (Figure 2).

It has been reported that the activation of JNK and ERK is involved in flolitin-1 expression. Therefore, we measured the levels of phosphorylated c-Jun N-terminal kinase (p-JNK) and phosphorylated extracellular signal-regulated kinase (p-ERK) using Western blotting. Data showed that p-JNK levels in the salivary glands were significantly elevated in the 9-month-old APP-KI group compared with the 2-month-old APP-KI and WT groups. The 9-month-old WT group also showed significantly higher p-JNK levels than the 2-month-old APP-KI and WT groups (Figure 3).

Similarly, p-ERK levels in the salivary gland were significantly increased in the 9-month-old APP-KI group compared with the 2-month-old APP-KI and WT groups, as well as the 9-month-old WT group. The 9-month-old WT group also exhibited significantly higher p-ERK levels than the 2-month-old APP-KI and WT groups (Figure 4).

Finally, salivary gland Aβ_42_ levels were significantly elevated in the 9-month-old APP-KI group compared with the 2-month-old APP-KI and WT groups, as well as the 9-month-old WT group (Figure 5).

## 4. Discussion

In this study, we investigated whether salivary flotillin-1 could serve as a potential diagnostic biomarker for AD by analyzing flotillin-1 levels in the saliva and salivary glands of APP-KI mice and examining the underlying regulatory mechanisms.

AD is associated with progressive accumulation of Aβ in the brain, and APP-KI mice have been reported to exhibit age-dependent increases in cerebral Aβ deposition. In this study, 2-month-old APP-KI mice were considered a pre-symptomatic model, whereas 9-month-old APP-KI mice were regarded as an AD pathology model [6].

Our findings demonstrated that salivary flotillin-1 levels increased significantly with age in WT and APP-KI mice; the 9-month-old WT group showed higher levels than the 2-month-old WT group, and the 9-month-old APP-KI group showed higher levels than the 2-month-old APP-KI group. Importantly, flotillin-1 levels in the 9-month-old APP-KI group were significantly higher than those in age-matched WT mice. These results suggest that flotillin-1 levels were increased with aging regardless of AD pathology; however, AD pathology further amplifies this increase, supporting the potential utility of salivary flotillin-1 as a potential biomarker for AD. Because salivary protein levels could be affected by secretion volume or total protein concentration, these factors were assessed to avoid misinterpretation. Consistent with the findings of previous reports [6,15], our results showed no significant differences between APP-KI and WT mice in salivary secretion volume or total protein concentration. Thus, the increase in flotillin-1 observed in 9-month-old APP-KI mice is unlikely to be attributed to differences in saliva volume or total protein, but rather to AD-related alterations in flotillin-1 metabolism in the saliva and salivary glands. Interestingly, these findings contrast with prior studies reporting reduced flotillin-1 levels in the serum and CSF of patients with AD [11]. This discrepancy may reflect differences in distribution dynamics, secretion mechanisms, or cell-type specificity among body fluids. For example, reduced flotillin-1 levels in serum and CSF may result from intracellular accumulation of flotillin-1 in the brain or from decreased exosome release due to Aβ-mediated JNK inhibition [3,11,16,17]. Conversely, saliva is secreted through exocytosis from the salivary glands; thus, protein levels in saliva may reflect protein production within these glands. Moreover, salivary Aβ concentrations increase in mice with Aβ accumulation in the salivary glands [9]. Although this differs from the behavior of Aβ in serum and CSF, it supports the idea that enhanced protein expression in the salivary glands can be reflected in saliva. Similarly, the present study demonstrated increased flotillin-1 levels in the salivary glands of 9-month-old WT mice compared with 2-month-old WT mice, and significantly higher levels in 9-month-old APP-KI mice than in 9-month-old WT mice. On the other hand, the concentrations of flotillin-1 in CSF and blood may remain low. One possible explanation is that aging- and AD-related cellular stress could activate signaling pathways, leading to upregulation of flotillin-1 expression. In addition, salivary glands are under strong autonomic regulation, and age-related changes in glandular function and vesicular secretion may elevate salivary flotillin-1 levels independently of systemic concentrations [18,19]. These findings suggest that salivary flotillin-1 may reflect local cellular processes associated with vesicular transport or AD-specific secretory mechanisms, highlighting its potential utility as a non-invasive biomarker.

These findings suggest that upregulation of flotillin-1 in the salivary glands contributes to elevated salivary flotillin-1 levels and may underlie the contrasting results relative to serum and CSF.

Furthermore, we examined the intracellular signaling mechanisms underlying the increase in flotillin-1 observed in the salivary glands. Recent studies have reported that flotillin-1 expression is regulated, at least in part, by the MAPK signaling pathway, particularly the JNK and ERK cascades [18,19]. In the present study, p-JNK and p-ERK levels were significantly higher in 9-month-old WT mice than in 2-month-old WT mice, and in 9-month-old APP-KI mice than in 2-month-old APP-KI mice, suggesting that aging activates JNK and ERK. Similar age-related activation of ERK has been reported in vascular endothelial cells and cardiomyocytes, where ERK activation is enhanced by oxidative stress responses and inflammation-associated signaling [20,21]. Age-dependent activation of JNK has also been described in the kidney and atrium [14,22,23]. Accordingly, the JNK and ERK activation observed in salivary glands in this study is likely attributable, at least partially, to aging. Notably, p-ERK levels were significantly higher in 9-month-old APP-KI mice than in age-matched WT mice, whereas p-JNK levels did not differ significantly. This finding suggests that in salivary glands under AD pathology, selective activation of ERK occurs. Furthermore, Aβ stimulation activates ERK signaling in the brain, contributing to increased flotillin-1 levels [24,25]. Consistent with this, our results suggest that accumulation of Aβ in salivary glands excessively activates the ERK pathway, thereby promoting flotillin-1 expression. Although immunohistochemical studies have previously reported Aβ accumulation in the salivary glands of APP-KI mice, the present study is the first to biochemically confirm elevated salivary gland Aβ levels using ELISA. Collectively, our findings suggest that Aβ accumulation in salivary glands enhances ERK activation, which in turn promotes flotillin-1 expression in the glands, ultimately leading to increased secretion of flotillin-1 into saliva.

Previous studies have indicated that salivary Aβ is not a reliable biomarker, largely because its concentration is extremely low and highly susceptible to environmental and physiological fluctuations, resulting in poor analytical stability and reproducibility. In contrast, flotillin-1 may offer a more robust alternative [26]. Given its higher detectability, stronger association with vesicular trafficking, and potentially closer link to AD-related cellular processes, salivary flotillin-1 may provide greater stability as a peripheral biomarker compared with salivary Aβ.

In this study, we demonstrated that flotillin-1 levels were nearly doubled in 9-month-old APP-KI mice compared with control mice. These findings suggest that salivary flotillin-1 may increase in the presence of cerebral Aβ accumulation, as observed in this study. However, this study was conducted using a mouse model, and the results may not necessarily be directly applicable to humans, which represents a limitation of our experimental design.

To overcome this limitation, future studies are needed to examine human saliva samples to clarify the relationship between salivary flotillin-1 levels and brain Aβ burden in patients with Alzheimer’s disease. In addition, we also found that flotillin-1 levels increase with age even in wild-type mice, it is needed to establish age-specific reference values by determining mean levels across age groups will enable assessments that account for age-related changes. If these issues are addressed, flotillin-1 has the potential to be developed as a non-invasive diagnostic indicator in the future.

## 5. Conclusions

This study demonstrated that Aβ accumulation in salivary glands under AD pathology may activate the ERK cascade, thereby promoting flotillin-1 expression and contributing to elevated salivary flotillin-1 levels. These findings suggest that flotillin-1, a molecule influenced by AD pathology, may serve as a noninvasive biomarker when measured in saliva.

## Figures and Tables

**Figure 1 diagnostics-16-00061-f001:**
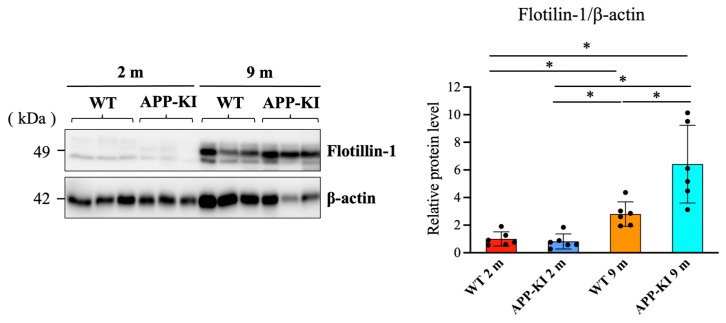
Salivary flotillin-1 levels in amyloid precursor protein knock-in (APP-KI) and wild-type (WT) mice. Flotillin-1 levels in saliva were assessed using Western blotting. Data were analyzed using the Kruskal–Wallis test, followed by the Steel–Dwass post hoc test for multiple comparisons. Data are presented as mean ± SD. *n* = 6 per group. Statistical significance was set at * *p* < 0.05.

**Figure 2 diagnostics-16-00061-f002:**
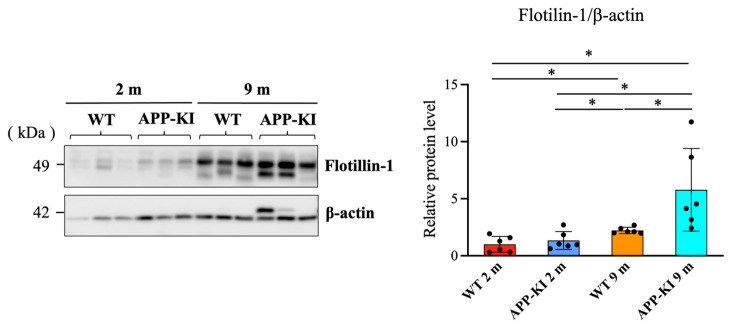
Flotillin-1 levels in the salivary glands of amyloid precursor protein knock-in (APP-KI) and wild-type (WT) mice. Flotillin-1 levels in the salivary glands were evaluated using Western blotting. Data were analyzed using the Kruskal–Wallis test, followed by the Steel–Dwass post hoc test for multiple comparisons. Data are presented as mean ± SD. *n* = 6 per group. Statistical significance was set at * *p* < 0.05.

**Figure 3 diagnostics-16-00061-f003:**
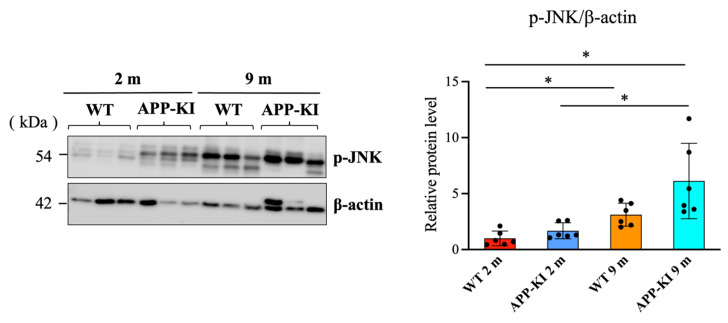
Phosphorylated c-Jun N-terminal kinase (p-JNK) levels in the salivary glands of amyloid precursor protein knock-in (APP-KI) and wild-type (WT) mice. p-JNK levels in the salivary glands were evaluated using Western blotting. Data were analyzed using the Kruskal–Wallis test, followed by the Steel–Dwass post hoc test for multiple comparisons. Data are presented as mean ± SD. *n* = 6 per group. Statistical significance was set at * *p* < 0.05.

**Figure 4 diagnostics-16-00061-f004:**
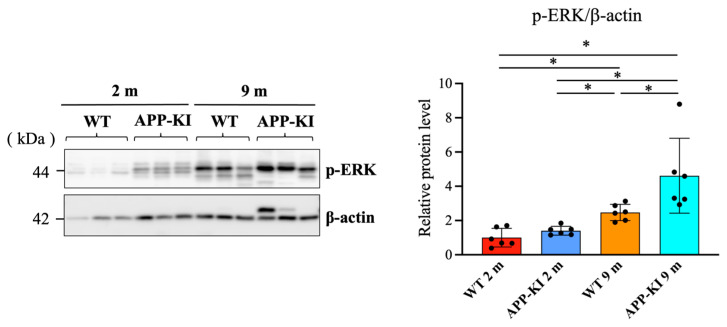
Phosphorylated extracellular signal-regulated kinase (p-ERK) levels in the salivary glands of amyloid precursor protein knock-in (APP-KI) and wild-type (WT) mice. p-ERK levels in the salivary glands were evaluated using Western blotting. Data were analyzed using the Kruskal–Wallis test, followed by the Steel–Dwass post hoc test for multiple comparisons. Data are presented as mean ± SD. *n* = 6 per group. Statistical significance was set at * *p* < 0.05.

**Figure 5 diagnostics-16-00061-f005:**
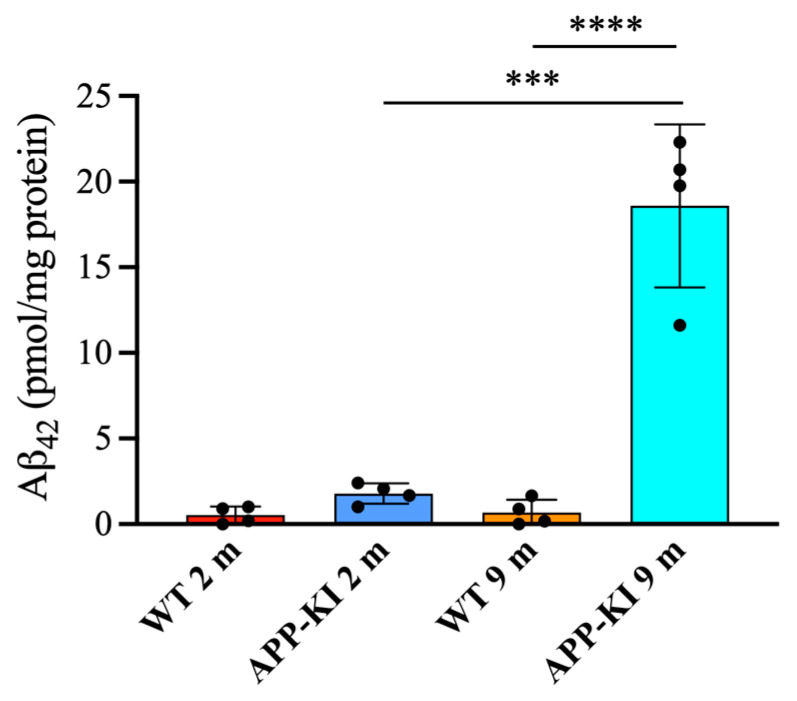
Amyloid beta 1–42 (Aβ_42_) levels in the salivary glands of amyloid precursor protein knock-in (APP-KI) and wild-type (WT) mice. Aβ_42_ levels in the salivary glands were evaluated using a Human Aβ_42_ ELISA kit. Data were analyzed using the Kruskal–Wallis test, followed by the Steel–Dwass post hoc test for multiple comparisons. Data are presented as mean ± SD. *n* = 4 per group. Statistical significance was set at *** *p* < 0.001, **** *p* < 0.0001.

## Data Availability

The datasets presented in this article are not readily available because the data are part of an ongoing study. Requests to access the datasets should be directed to Sunao Kawakami.

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
