# Peer review of "Elevated Flotillin-1 in Saliva and Salivary Glands: A Novel Non-Invasive Biomarker in an Alzheimer’s Disease Mouse Model"

_diagnostics, 2025, doi:10.3390/diagnostics16010061_

Round 1

Reviewer 1 Report

Comments and Suggestions for Authors

The study focus on investigate of the availability of AD biomarkers in saliva for less invasive specimen than the usual sample type. There is need in the literature for such study to show that AD biomarker could be found in saliva and this type of diagnostics is very convenient for both health care provider as well as for the patients especially for AD patients because it is aging is key player as it is one of the top risk factor of developing the disease. Collecting saliva sample is much easier from blood sample for elderly individuals. The manuscript is informative and may enrich the literature regarding Alzheimer’s disease. The study is valuable in term of innovation and development. The study used Western blot and ELISA to detect the accumulation of Amyloid Beta and fktillin-1 which those two assay are widely used in such detecting experiments which they usually give sold results.

  • If you already used brain tissues, it is good to mention in the manuscript to confirm the progression of the mice condition, should use neurons biomarkers to illustrate the condition of those neurons as mature and immature neurons to show the progression of the AD agent from brain tissue.
  • The AD knock in agent used in the study may show the symptom after certain age like 6 months. Therefore, explain more about the reason to use 2 month mouse model?
  • The older animal model used in the study equal less than half life in human, so why not to use 12-24 month mice unless the study objective is to preventative and classifying people with high risk of developing the disease.
  • Line 66 (Cerebral Aβ accumulation begins before the onset of 66 clinically detectable cognitive impairment) should add reference.
  • Should write more about Flotillin-1 association with other disease rather than just AD.
  • Line 89 (It is possible that flotillin-1 may be a potential biomarker for saliva) couldn’t understand.
  • Amyloid-β 42 has small molecular weight to run in western blot with PVDF membrane, but have you tried nitrocellulose membrane instead of using two techniques (western blot and ELISA)?
  • In line 181, (we investigated whether aging and AD pathologies can affect flotillin-1 levels), using the term (aging) should be for aged mouse and what have been used in this experiment considered as less than middle age mice.
  • Figure 7, need to be edited as no need for n.s. etc..
  • If the Flotillin-1 increase by age as written in line 246, how it could be marker of AD and in what level the AD could amplifies this increase?

Author Response

Reviewer 1

  • Comment 1:

If you already used brain tissues, it is good to mention in the manuscript to confirm the progression of the mice condition, should use neurons biomarkers to illustrate the condition of those neurons as mature and immature neurons to show the progression of the AD agent from brain tissue.

Response 1:

Thank you for your comment. As this study did not include the use of brain samples, we were unable to provide data addressing this aspect.

  • Comment 2:

The AD knock in agent used in the study may show the symptom after certain age like 6 months. Therefore, explain more about the reason to use 2 month mouse model?

Response 2:

Thank you for your comment. Because this new type of model mouse, APP-KI, begins Ab deposition and memory deficit around 3-4 month of age. The 2-month-old mice used in this study were APP-KI mice without cerebral Aβ accumulation and without cognitive symptoms, and thus they were used as the control group for the 9-month-old APP-KI mice.

  • Comment 3:

The older animal model used in the study equal less than half life in human, so why not to use 12-24 month mice unless the study objective is to preventative and classifying people with high risk of developing the disease.

Response 3:

Thank you for your insightful comment. We employed a mouse model that exhibits early age-dependent accumulation of brain Aβ, as I mentioned in our Response 2, resulting in cognitive decline. By 9 months of age, APP-KI mice show saturated levels of Aβ deposition in the brain along with observable cognitive symptoms. Therefore, our study was conducted using the 9-month-old mice, instead of using 12–24-month mice.

  • Comment 4:

Line 66 (Cerebral Aβ accumulation begins before the onset of 66 clinically detectable cognitive impairment) should add reference.

Response 4:

Taking your suggestion into account, we have added the relevant references accordingly. (added in the Introduction section, pp. 68, in the References section, pp. 363-5)

  • Comment 5:

Should write more about Flotillin-1 association with other disease rather than just AD.

Response 5:

We appreciate the reviewer’s valuable comment. Since our study did not investigate other types of dementia, because we had no appropriate model mice for other types of disease-causing dementia. Thus, we could not draw conclusions regarding their relationship with salivary flotillin-1 levels. However, your comment is very important for developing diagnostic marker for Alzheimer’s disease, and we will address this point in the future study. Anyway, based on current knowledge, flotillin dynamics have been shown to change in cultured cells in response to Aβ treatment, and alterations in blood flotillin levels have also been reported in patients with Alzheimer’s disease. Given that flotillin levels are known to be affected in the blood of individuals with Alzheimer’s disease, we focused our study specifically on AD.

  • Comment 6:

Line 89 (It is possible that flotillin-1 may be a potential biomarker for saliva) couldn’t understand.

Response 6: 

We apologize for any lack of clarity in the original text. We have totally revised the relevant section of the manuscript to make our intention more explicit by replacement of this sentence to a new one. (added in the Introduction section, pp. 90-95)

  • Comment 7:

Amyloid-β 42 has small molecular weight to run in western blot with PVDF membrane, but have you tried nitrocellulose membrane instead of using two techniques (western blot and ELISA)?

Response 7: 

Thank you very much for this valuable comment. We did not use a nitrocellulose membrane in our experiments. As you suggested, it is possible that the signal might have been detectable with a nitrocellulose membrane. In this experiment, therefore, instead of western blot analysis, we employed ELISA, which allows quantitative measurement, to detect flotillin-1 levels.

  • Comment 8:

In line 181, (we investigated whether aging and AD pathologies can affect flotillin-1 levels), using the term (aging) should be for aged mouse and what have been used in this experiment considered as less than middle age mice.

Response 8: 

You are absolutely right. The term we used was inappropriate, and we have revised the manuscript accordingly. Thank you for pointing this out. (added in the Results section, pp. 178-179)

  • Comment 9:

Figure 7, need to be edited as no need for n.s. etc..

Response 9: 

We appreciate your comment and have revised the manuscript accordingly.

  • Comment 10:

If the Flotillin-1 increase by age as written in line 246, how it could be marker of AD and in what level the AD could amplifies this increase?

Response 10: 

We appreciate your valuable comment. In this study, we demonstrated that the concentration of flotillin-1 in 9-month-old APP-KI mice was nearly doubled compared to wild-type control mice at the same age. These results suggest that salivary flotillin-1 levels increase in the presence of cerebral Aβ accumulation. However, an age-related increase was also observed. Therefore, it is needed to establish reference values based on the mean values of each age group. And then, we can evaluate the flotillin-1 levels with the age-matched reference values of flotillin-1 levels.

A discussion addressing the issue you mentioned has been added to the manuscript. (added in the discussion section, pp. 314-320)

Reviewer 2 Report

Comments and Suggestions for Authors

The present work assessed flotillin-1 levels in saliva and salivary 98 glands and their alterations under AD pathologies.

Furthermore, the authors explored the mechanisms regulating these changes using an AD mouse model analyzing phosphorylated c-Jun N-terminal kinase and extracellular signal-regulated kinase levels.

Some comments could be taken into account for improvement.

the selection between young and old mice, however 9 months are enough to investigate significant alterations in AD pathology, especially in the wild type organisms.

The authors used six male mice, wild type and APP-KI mice. Can the authors explain in details the selection of the age of 2 months and 9 months. I can understand

Salivary secretion levels and total protein concentration could be transferred in supplementary material as no significant changes were found.

The authors measured amyloid beta levels though ELISA using a human Aβ42 kit. However, the study used animal models. Furthermore, Figure 7 talks about western blotting results. This point should be revised.

Limitations of the present process and some future directions could be also discussed.

Author Response

Reviewer 2

  • Comment 1:

The authors used six male mice, wild type and APP-KI mice. Can the authors explain in details the selection of the age of 2 months and 9 months. I can understand

Response 1:

We used an APP-KI mouse model in which cerebral Aβ accumulation begins early in life and subsequently leads to cognitive impairment with aging. In this APP-KI mice, Ab deposition and memory impairment begins around 3-4 month of age and reaches their plateau around 8-9 month of age. Therefore, 2-month-old APP-KI mice do not exhibit Aβ deposition or cognitive symptoms, whereas 9-month-old APP-KI mice show clear Aβ accumulation and cognitive deficits.

  • Comment 2:

Salivary secretion levels and total protein concentration could be transferred in supplementary material as no significant changes were found.

Response 2:

Taking your comment into account, Figure 1 and Figure 2 have been moved to the Supplementary Material. We have also revised the corresponding descriptions in the main text and updated the figure citations to “Supplementary Figure A and B” to maintain consistency.

  • Comment 3:

The authors measured amyloid beta levels though ELISA using a human Aβ42 kit. However, the study used animal models. Furthermore, Figure 7 talks about western blotting results. This point should be revised.

Response 3:

The reviewer’s comments are reasonable. The APP-KI mouse model used in our study carries a humanized Aβ sequence (including the Swedish/Iberian mutations) and therefore produces human Aβ40/42 peptides. Therefor we used ELISA kit specific for human Aβ42.

Therefore, using a human Aβ42 ELISA kit is scientifically valid and consistent with the literature.

We also note and have corrected an error in the manuscript: the results presented in Figure 7 are from ELISA measurements, not from western blotting. The figure legend and relevant text have been updated accordingly to avoid any further confusion.

  • Comments 4:

Limitations of the present process and some future directions could be also discussed.

Response 4:

We appreciate your valuable comment. A discussion addressing the issue you mentioned has been added to the manuscript. (added in the discussion section, pp. 307-320)

“[In this study, we demonstrated that flotillin-1 levels were nearly doubled in 9-month-old APP-KI mice compared with control mice. These findings suggest that salivary flotillin-1 may increase in the presence of cerebral Aβ accumulation, as observed in this study.
However, this study was conducted using a mouse model, and the results may not necessarily be directly applicable to humans, which represents a limitation of our experimental design. To overcome this limitation, future studies are needed to examine human saliva samples to clarify the relationship between salivary flotillin-1 levels and brain Aβ burden in patients with Alzheimer’s disease. In addition, we also found that flotillin-1 levels increase with age even in wild-type mice, it is needed to establish age-specific reference values by determining mean levels across age groups will enable assessments that account for age-related changes. If these issues are addressed, flotillin-1 has the potential to be developed as a non-invasive diagnostic indicator in the future. ]”

Reviewer 3 Report

Comments and Suggestions for Authors

The manuscript "Flotillin-1 is increased in the saliva and salivary glands of Alzheimer's disease model mice" presents compelling evidence that Flotillin-1—a key regulator of cellular vesicular trafficking—is significantly elevated in the saliva and salivary glands of APP-KI mice, an established Alzheimer's disease (AD) model. The authors also measured Aβ1-42 peptides in saliva, though no consistent pattern emerged. Detection relied on robust methods including ELISA and Western blotting to assess Flotillin-1 and related pathways. With the pressing need for non-invasive, early AD biomarkers, this work meaningfully advances saliva-based diagnostics, a promising frontier supported by emerging literature [PMID: 32225073].

To strengthen the manuscript, I offer the following constructive suggestions:

  1. Title Revision: The current title could better highlight Flotillin-1's potential as a novel salivary biomarker. Consider: "Elevated Flotillin-1 in Saliva and Salivary Glands: A Novel Non-Invasive Biomarker in an Alzheimer's Disease Mouse Model."

  2. Figure Legends - Enhanced Statistics: Include specific statistical details (e.g., test type, post-hoc analysis, p-values) in legends for Figures 1 and 2. Combining these similar figures would improve visual flow and readability.

  3. Supplemental Analysis: Add a direct t-test comparison between WT 9m and APP-KI 9m groups in the supplements to bolster Figures 3 and 4.

  4. Comparative Fluid Measurements: Did the authors assess Flotillin-1 and Aβ1-42/1-40 levels in blood or CSF? This would validate model consistency with human AD literature and address inter-species/inter-fluid variability—a critical validation step.

  5. Data Visualization: Replace bar charts with dot plots (or violin plots) to display individual data points, enabling readers to assess variability and outliers directly.

  6. Figure Legend Clarity: Legends should focus strictly on methods, sample sizes, and statistics, avoiding result interpretation. Consolidating related figures would reduce fragmentation and enhance focus.

  7. Mechanistic Discussion: Why does Flotillin-1 increase with age in mice? What explains its elevation in saliva despite potentially lower levels in CSF/blood? These patterns warrant deeper exploration in the Discussion, linking to vesicular trafficking or AD-specific secretion [PMID: 32225073].

  8. Housekeeping Gene Selection: The chosen housekeeping gene appears inconsistent across samples. Validate or replace it with a more stable reference (e.g., GAPDH or β-actin, confirmed via geNorm or NormFinder).

  9. p-JNK/p-ERK Consistency: Elevated p-JNK and p-ERK levels are reported, but discrepancies exist between abstract, text, and graphs. The Introduction should justify their measurement (e.g., roles in neuroinflammation/APP processing) and align all descriptions.

  10. Aβ Findings and Biomarker Comparison: The marked Aβ elevation raises questions—is Aβ a superior biomarker to Flotillin-1? Fully discuss these results, including saliva-blood-CSF discrepancies and ERK/JNK inconsistencies, with literature context on salivary Aβ variability.

Author Response

Reviewer 3

  • Comments 1:

Title Revision: The current title could better highlight Flotillin-1's potential as a novel salivary biomarker. Consider: "Elevated Flotillin-1 in Saliva and Salivary Glands: A Novel Non-Invasive Biomarker in an Alzheimer's Disease Mouse Model."

Response 1:

Thank you very much for your valuable advice. Based on your suggestion, we have changed the title as recommended.

  • Comment 2:

Figure Legends - Enhanced Statistics: Include specific statistical details (e.g., test type, post-hoc analysis, p-values) in legends for Figures 1 and 2. Combining these similar figures would improve visual flow and readability.

Response 2:

Thank you very much for your insightful suggestion. We have adjusted the figure legends as advised. Moreover, since other reviewer also recommended moving Figures 1 and 2 to the Supplementary Material, we have implemented this modification accordingly.

  • Comments 3:

Supplemental Analysis: Add a direct t-test comparison between WT 9m and APP-KI 9m groups in the supplements to bolster Figures 3 and 4.

Response 3:

Thank you very much for your insightful suggestion. At this stage, since significant differences have already been observed using the Kruskal–Wallis multiple comparisons, we would like to continue with the current approach. We hope this is acceptable.

  • Comments 4:

Comparative Fluid Measurements: Did the authors assess Flotillin-1 and Aβ1-42/1-40 levels in blood or CSF? This would validate model consistency with human AD literature and address inter-species/inter-fluid variability—a critical validation step.

Response 4:

We totally agree with your suggestion. Unfortunately, however, we did not have blood samples nor the necessary material at present to conduct this analysis. We plan to include this aspect in our study design when we move on to investigations using human saliva samples in the future.

  • Comments 5:

Data Visualization: Replace bar charts with dot plots (or violin plots) to display individual data points, enabling readers to assess variability and outliers directly.

Response 5:

Thank you very much for your valuable suggestion. We have updated the figure as recommended.

  • Comments 6:

Figure Legend Clarity: Legends should focus strictly on methods, sample sizes, and statistics, avoiding result interpretation. Consolidating related figures would reduce fragmentation and enhance focus.

Response 6:

Thank you very much for your valuable suggestions. In response, we have incorporated the statistical details into the Figure Legends and removed the interpretative descriptions. Moreover, we have merged Figures 1 and 2 and transferred the unified figure to the Supplementary Material.

  • Comments 7:

Mechanistic Discussion: Why does Flotillin-1 increase with age in mice? What explains its elevation in saliva despite potentially lower levels in CSF/blood? These patterns warrant deeper exploration in the Discussion, linking to vesicular trafficking or AD-specific secretion [PMID: 32225073].

Response 7:

The reviewer’s comment is reasonable and important to be discussed. The underlying reason remains unclear at this stage. However, we have added description regarding this point in the Discussion section. (added in the discussion section, pp. 267-273,322-335)

  • Comment 8:

Housekeeping Gene Selection: The chosen housekeeping gene appears inconsistent across samples. Validate or replace it with a more stable reference (e.g., GAPDH or β-actin, confirmed via geNorm or NormFinder).

Response 8:

Thank you for your comment. Figure 7 was mistakenly described as a western blot analysis, but it was actually an ELISA, and we have corrected this error. With this clarification, the housekeeping gene can now be consistently presented as β-actin across the figures.

  • Comment 9:

p-JNK/p-ERK Consistency: Elevated p-JNK and p-ERK levels are reported, but discrepancies exist between abstract, text, and graphs. The Introduction should justify their measurement (e.g., roles in neuroinflammation/APP processing) and align all descriptions.

Response 9:

Thank you very much for your valuable suggestion. In response, we have made revisions to both the Introduction and the Figure Legends (added in the Introduction section, pp. 97-104).

“[Previous studies have demonstrated that the MAPK pathways regulate flotillin-1 expression in several cell types, including neurons, macrophages, and epithelial cells. In these cells, Aβ-induced activation of the JNK and ERK pathways modulates flotillin-1 expression and membrane microdomain dynamics. Because JNK is primarily involved in stress- and inflammation-related signaling, and ERK plays a key role in cell survival and membrane trafficking, we further examined JNK/ERK activation and Aβ levels in the salivary glands in addition to measuring salivary and glandular flotillin-1 levels.]”

  • Comment 10:

Aβ Findings and Biomarker Comparison: The marked Aβ elevation raises questions—is Aβ a superior biomarker to Flotillin-1? Fully discuss these results, including saliva-blood-CSF discrepancies and ERK/JNK inconsistencies, with literature context on salivary Aβ variability.

Response 10:
Thank you very much for your insightful and constructive comment. In accordance with your suggestion, we have substantially expanded our discussion regarding the interpretation of Aβ findings, the comparison between Aβ and flotillin-1 as potential biomarkers, and the discrepancies observed across saliva, blood, and CSF.

Although salivary Aβ has been investigated in previous studies, it has not been established as a reliable biomarker. This is mainly due to its extremely low concentration in saliva and its high susceptibility to environmental and physiological fluctuations, resulting in poor analytical stability and reproducibility. In contrast, flotillin-1 may offer a more robust alternative because of its higher detectability, stronger association with vesicular trafficking, and closer link to AD-related cellular processes. We have incorporated this discussion into the revised manuscript, together with relevant literature on the variability and methodological limitations of salivary Aβ measurements (added in the discussion section, pp. 267-273,322-335)

We have also strengthened the discussion on the discrepancies observed between saliva, blood, and CSF. Specifically, flotillin-1 elevation was clearly observed in the salivary glands, while peripheral or central dynamics do not necessarily follow the same pattern. We discuss that these differences may reflect local cellular regulation, autonomic control of the salivary glands, and tissue-specific stress responses.

Furthermore, based on previous reports that MAPK pathways (ERK/JNK) regulate flotillin-1 expression in multiple cell types, we examined ERK and JNK activation in the salivary glands. In the revised manuscript, it is clarified that ERK activation was consistent with the increase in flotillin-1, whereas JNK did not show a similar pattern, suggesting that this inconsistency may reflect pathway-specific regulatory mechanisms.

All of these revisions have been added to the updated Discussion section (added in the discussion section, pp. 267-273,322-335) and are highlighted in the re-submitted manuscript. We sincerely appreciate your valuable comment, which has improved the clarity and scientific depth of our study.

“[On the other hand, the concentrations of flotillin-1 in CSF and blood may remain low. One possible explanation is that aging- and AD-related cellular stress could activate signaling pathways, leading to upregulation of flotillin-1 expression. In addition, salivary glands are under strong autonomic regulation, and age-related changes in glandular function and vesicular secretion may elevate salivary flotillin-1 levels independently of systemic concentrations. These findings suggest that salivary flotillin-1 may reflect local cellular processes associated with vesicular transport or AD-specific secretory mechanisms, highlighting its potential utility as a non-invasive biomarker. ]”

“[Previous studies have indicated that salivary Aβ is not a reliable biomarker, largely because its concentration is extremely low and highly susceptible to environ-mental and physiological fluctuations, resulting in poor analytical stability and repro-ducibility. In contrast, flotillin-1 may offer a more robust alternative. Given its higher detectability, stronger association with vesicular trafficking, and potentially closer link to AD-related cellular processes, salivary flotillin-1 may provide greater stability as a peripheral biomarker compared with salivary Aβ. ]”

Round 2

Reviewer 3 Report

Comments and Suggestions for Authors

NA